# KATP Channel Inhibitors Reduce Cell Proliferation Through Upregulation of H3K27ac in Diffuse Intrinsic Pontine Glioma: A Functional Expression Investigation

**DOI:** 10.3390/cancers17030358

**Published:** 2025-01-22

**Authors:** Marina Antonacci, Fatima Maqoud, Annamaria Di Turi, Morena Miciaccia, Maria Grazia Perrone, Antonio Scilimati, Domenico Tricarico

**Affiliations:** 1Department of Pharmacy-Pharmaceutical Sciences, University of Bari “Aldo Moro”, 70125 Bari, Italy; marina.antonacci@uniba.it (M.A.); f.maqoud@gmail.com (F.M.); annamariadituri@gmail.com (A.D.T.); morena.miciaccia@uniba.it (M.M.); mariagrazia.perrone@uniba.it (M.G.P.); 2Functional Gastrointestinal Disorders Research Group, National Institute of Gastroenterology Saverio de Bellis, I.R.C.C.S. Research Hospital, 70013 Castellana Grotte, Italy

**Keywords:** diffuse intrinsic pontine glioma, TRPV1 and KATP channels, cell proliferation, patch-clamp, gene expression

## Abstract

The Diffuse intrinsic pontine glioma (DIPG) is a rare aggressive fatal pediatric brain tumor with no effective therapy. The authors aim is to investigate the role of the subunits of the KATP and TRPV1 ion channels as novel alternative targets regulating cell proliferation in DIPG. We found that the KATP channel inhibitors glibenclamide and repaglinide prescribed in the hyperglycemic conditions including the Neonatal Diabetes (glibenclamide-AMGLIDIA^®^), were the most effective and potent antiproliferative drugs in two human DIPG cell lines used as cell models of the aggressive DIPG. These findings can be of advantage in a combination therapy with novel drug class such as the Pi3-kinase inhibitors that show hyperglycemia as a side effect. The Pi3-kinase inhibitor paxalisib has been successfully proposed in DIPG patients but the observed hyperglycemia limited its use. The combination therapy of KATP channel inhibitors as oral therapy with Pi3-kinase inhibitors may have synergic antiproliferative effects controlling the hyperglycemia.

## 1. Introduction

Diffuse midline glioma (DMG) is a fatal childhood high-grade glioma (pHGG), and its treatment is still a global challenge [1,2]. Diffuse intrinsic pontine glioma (DIPG), a subtype of DMG, is in the pons and carries a particularly poor prognosis, with a median survival of 1 year after the diagnosis. Radiotherapy provides relief from symptoms. Integration of systemic chemotherapy into the treatment protocol has consistently failed to demonstrate clinical benefit [3]. Moreover, surgery remains particularly challenging due to the anatomical localization of the tumor and its infiltrative growth. In recent decades, distinct pHGG entities have been identified based on recurrent epigenetic alterations. One of these DIPG subtypes, on which our study was conducted, is characterized by a post-translational modification, particularly a substitution of a lysine with a methionine at position 27 of the N-terminal tail (H3K27M) of histone 3 isoform 1 (H3.1) or histone 3 isoform 3 (H3.3). H3.1K27M tumors are like anaplastic astrocytoma on histopathology and tend to occur in younger patients; instead, H3.3K27M tumors are associated with TP53 and ATRX mutations and more frequently resemble glioblastoma [4,5,6,7]. H3.1K27M is identified in 12–20% of DIPG cases, with a median survival rate of 12–15 months, while H3.3K27M is identified in 65% of cases, with a median overall survival of 9–12 months. Pharmacological investigations are conducted on two patient-derived DIPG cell cultures (DIPG-36 carrying the H3.1K27M and DIPG-50 carrying the H3.3K27M), provided and previously characterized by Dr Michelle Monje with approval of the Institutional Review Board (Stanford University). Being the predominant molecular profile, these DIPG cell lines derived from two patients were selected for the study, encouraging the exploration of new therapeutic strategies in DIPG.

The ion channel’s role in cell division and tumor biology has been increasingly proved [8,9,10,11]. Few studies have specifically investigated genes encoding ion channels and transporters and how relevant they are in pediatric gliomas, even if these proteins participate in brain functioning (neuronal and glial) and are dysregulated in brain tumors [12,13,14,15]. The reason for such a small number of studies is certainly the low incidence of the disease and the rarity of the tissues appropriate for molecular characterization [16,17]. Anyway, these studies are crucial because in pediatric brain tumors, ion/transporter channels are upregulated in 33% and downregulated in 48% of cases [18].

ATP-dependent potassium channels (KATP) opening is regulated by intracellular ATP’s concentration variations. They are hetero-multimeric complexes consisting of SUR accessory subunits (SUR1-SUR2A–SUR2B) and main subunits (Kir6.1 and Kir6.2) forming the central pore. They are ubiquitous and present in all tissues [19]. Specifically, the mitochondrial and surface KATP channels are proposed as targets in cancers [20]. They are a well-known target for sulfonylureas and glinides drugs in diabetes and muscle atrophy [21], and for cardiovascular drugs [22,23]. KATP channels have been shown to also be regulated by H_2_S modulators [24] and engage in neurodegeneration and channelopathies [25,26]. There is recent, emerging evidence of their expression in a variety of cancer cell lines like human gastric cancer, human bladder cancer, and glioma, [27,28] as well as their role in the cancer progression [29].

Transient Receptor Potential (TRP) channels are a family of non-cation-selective ion channels, activated by various stimuli such as temperature, pH, oxidative stress, heat, and bacterial toxins. Pan cancer analysis recently associated TRP channels with cancer [30]. The Transient Receptor Potential Vanilloid Receptor 1 (TRPV1) was the first one to be identified and is now the most studied channel of this family. TRPV1 is found to be expressed in a wide range of tissues such as skin, gastrointestinal tract, urinary epithelial cells, pancreatic cells, and immune cells. The involvement in neuropathic pain, neurogenic inflammation, autoimmune disease, cancer, and immune cell functioning of these channels was also recently emphasized [31]. Also, the TRPV1 channel has recently been associated with the proliferation of osteoblastic cells [32].

In this study, DIPG-36 and DIPG-50 cell lines were used to screen KATP and TRPV1 channels modulators to verify which one had the greatest antiproliferative effect. Repaglinide and glibenclamide were identified; and to a lesser extent, capsaicin and capsazepine, TRPV1 agonist and antagonist, respectively, behaved at higher concentrations. The effects of the ion channel modulators were compared with those of chemotherapeutics, such as cisplatin, an alkylating agent binding to DNA through the formation of cross-links between complementary strands, able to interfere with all phases of the cell cycle, and doxorubicin, an antimitotic with binding cellular DNA that inhibits nucleic acid synthesis and mitosis, causing chromosomal aberrations. Doxorubicin improves the prognosis of children suffering from neuroblastoma and it is also used in DIPG [33].

## 2. Materials and Methods

*Cell culture.* Patient-derived diffuse intrinsic pontine glioma DIPG-36, DIPG-50 cells were provided by Prof. Michelle Monje (Stanford University) and cultured according to her protocol. As the predominant molecular profile, these two patient-derived DIPG cell lines were selected for the study, encouraging the exploration of new therapeutic strategies in DIPG. These cells were maintained in Tumor Stem Media (TSM) consisting of a 1:1 mixture of DMEM/F12 and Neurobasal (-A) (Invitrogen, Thermo Fisher Scientific Inc., Waltham, MA, USA), supplemented with B27 (-A) (Life Technologies, Thermo Fisher Scientific Inc., Waltham, MA, USA), 20 ng/mL human basic fibroblast growth factor (bFGF) (Life Technologies), 20 ng/mL recombinant human epidermal growth factor (EGF) (Life Technologies), 10 ng/mL platelet-derived growth factor-AA (PDGF-AA), 10 ng/mL platelet-derived growth factor-BB (PDGF-BB) (Life Technologies) and 20 ng/mL heparin (StemCell Technologies) at 37 °C in 5% CO_2_. DIPG-36 cell line grew in adhesion, as a monolayer, and passages between P32 and P36 were used; instead DIPG-50 cell line grew in suspension and in adhesion as well, and passages between P25 and P30 were used. Multi-well crystal violet staining assay: cells, previously counted using the Scepter 2.0 counter (Merck KGaA, Darmstadt, Germany), were plated in a 96 multi-well plate with a density of 8 × 10^3^ cells/well. After 24 h of incubation, different concentrations of drugs were added. At 6, 48, and 72 h incubation times, the medium was removed, and cells were fixed with 10% buffered formalin for 20 min at room temperature and subsequently stained using a 1% solution *v/v* of crystal violet for 30 min. Plates were washed with distilled water to remove the excess dye. Finally, acetic acid was added to elute the dye, and the absorbance at λ = 560 nm was read using the VictorTM spectrophotometer (Perkin Elmer, Waltham, MA, USA). Each experimental condition was accomplished in triplicate at least.

Murine bone marrow cells were prepared as previously described [34].

*CCK-8 intracellular dehydrogenase assay.* The activity of the intracellular dehydrogenases was evaluated by using the Cell Counting Kit-8 (CCK-8) (Sigma-Aldrich, Merck KGaA, Darmstadt, Germany), a sensitive test that utilizes water-soluble tetrazolium salt. WST-8 2-(2-methoxy-4-nitrophenyl)-3-(4-nitrophenyl)-5-(2,4-disulfophenyl)-2H-tetrazolium monosodium salt can be reduced by dehydrogenases in cells, producing an orange-colored formazan dye soluble in the culture medium. The cells were counted by using Scepter™ 2.0 (Merck Millipore Corporation, New York, NY, USA) cell counter, seeded in 96-well plates at a density of 8 × 10^3^ cells/well, and pre-incubated for 24 h under standard conditions. Then, cells were treated for 48 h and 72 h with different concentrations of each drug solution. Finally, the activity of the intracellular dehydrogenases was evaluated by adding CCK-8 solution (10 μL) to each well and measuring the absorbance at λ = 450 nm after 2 h of incubation with CCK-8 solution, using the microplate reader Victor^®^ 3V (PerkinElmer^®^, Waltham, MA, USA).

*Clonogenic assay.* 250 cells previously counted using the Scepter 2.0 counter (Merck Millipore Corporation) were plated before treatment in 60 mm plates. The cells were initially kept in the incubator for 24 h. Then, we evaluated the effect of the substances on the number of forming colonies after 72 h and 96 h of incubation. Each experimental condition was performed in triplicate. After the incubation, the culture medium with the cytotoxic compounds was removed and replaced with fresh medium. Cells were then kept in the culture for 2 weeks. At the end of the test, the colonies formed were fixed with 10% buffered formalin for 1 h at room temperature and stained with 0.05% *v/v* of crystal violet. In DIPG-50, Matrigel was added in advance, to promote cell adhesion due to the growth both in adhesion and in suspension of this cell type [35].

*Apoptosis assays*. Flow cytometry was performed using the CellEvent™ Caspase-3/7 Green Assay Kit C10427 (Thermo Fisher Scientific Inc,) in DIPG cells treated with REPA concentrations after 48 h of incubation. For cell volume analysis, the cells were sized using the Scepter™ 2.0 cell counter (Merck Millipore Corporation). Scepter™ cell counter measures volume, so it can quantify cells based on size and differentiates larger cells from smaller debris so that we could identify an effect of substance on cell diameter, concerning in a death pathway. Precisely, detailed cell volumes are drawn into the Scepter™ sensor. As cells flow through the aperture in the sensor, resistance increases. This increase in resistance causes a subsequent increase in voltage. Voltage changes are recorded as spikes with each passing cell, and these are proportional to the cell volume. The spikes of the same size are bucketed into a histogram and counted. This histogram gives quantitative data on cell morphology that can be used to examine the quality and health of the cell culture. The Scepter™ 2.0 cell counter (Merck Millipore Corporation) is compatible with 60 and 40 μm sensors; in our experiments we used the 60 μm sensor for particles between 6 and 36 μm.

*Whole-cell recordings.* Drug actions on the channel currents recorded during instantaneous I/V relationships were investigated by applying a depolarization protocol, in a range of potentials going from −100 to +180 mV (Vm) for DIPG-36 and DIPG-50 cells in 20 mV steps, starting from HP = −60 mV (Vm), in physiological conditions, with asymmetrical K+ ion concentrations (int K^+^: 132 × 10^−3^ M; ext. K^+^: 2.8 × 10^−3^ M) using the whole-cell patch clamp technique. All experiments were performed at room temperature (20–22 °C) and sampled at 2 kHz (filter = 1 kHz) using an Axopatch-1D amplifier equipped with a CV-4 head-stage (Axon Instruments, Foster City, CA, USA). Patch pipettes were pulled from PG52165-4. Patch Clamp #8250 Glass Capillaries, 1.65 mm o.d., 1.20 mm i.d. (World Precision Instruments, 175 Sarasota Center Blvd. Sarasota, FL, USA) with a vertical puller (PP-82 Narishige Tokyo, Japan) to give a resistance of 4–5 MW. Data acquisition and analysis was performed using pCLAMP 10 software suite (Axon Instruments, Foster City, CA, USA). Seal resistance was continuously monitored during the experiment; cells not showing a stable seal (R >> 1 GΩ) were not selected for further analysis.

*Drugs and solutions.* Whole-cell patch clamp experiments on DIPG-36 and DIPG-50 cells were performed in asymmetrical K^+^ ion concentrations. The pipette solution contained:132 mM K^+^-glutamate, 1 mM ethylene glycol-bis(β-aminoethylether)-*N*,*N*,*N*′,*N*′-tetraacetic acid (EGTA), 10 mM NaCl, 2 mM MgCl_2_, 10 mM HEPES, 1 mM Na_2_ATP, 0.3 mM Na_2_GDP (pH = 7.2). The bath solution contained 142 mM NaCl, 2.8 mM KCl, 1 mM CaCl_2_, 1 mM MgCl_2_, 11 mM glucose, and 10 mM HEPES (pH = 7.4). Substances used for patch clamp recordings or in vitro assays were: glibenclamide (GLIB), glimepiride (GLIMP), tolbutamide (TOLB), repaglinide (REPA), diazoxide (DIAZO), tetraethylammonium hydrochloride (TEA), BaCl_2_, capsaicin (CAPS), capsazepine (CAPSZ), cisplatin (CISP), and doxorubicin (DOXO) purchased from Sigma (Sigma-Aldrich, Merck KGaA, Darmstadt, Germany). Stock solutions of GLIB, GLIMP, REPA, DIAZO, DOXO, CISP, TEA, BaCl_2_, and CAPSZ were prepared by dissolving each drug in dimethyl sulfoxide (DMSO), whereas CAPS and TOLB were dissolved in ethanol. Stock solutions were stored in the refrigerator. Dilute solutions were prepared on the day of the experiments and maintained at room temperature. For cell proliferation assays, diluted solutions of drugs were prepared using the medium. For patch clamp experiments, microliter amounts of the stock solutions were added to the bath solutions as needed. DMSO did not exceed 0.07%,a concentration that does not normally affect current or cell proliferation.

*RT-Polymerase chain reaction.* The total RNA was extracted and purified from SU-DIPG-36 (DIPG-36) and SU-DIPG-50 (DIPG-50) cells using the RNA extraction kit LS1040 (Promega Corporation, Madison, WI, USA) according to the manufacturer’s protocol. The RNA quantity was measured with an ND-1000 NanoDrop spectrophotometer. Reverse transcription was conducted using GoScript™ Reverse Transcriptase A5001 (, Promega Corporation). Real-time PCR was performed in triplicate with the 7500 Fast Real-Time PCR System (Applied Biosystems, Thermo Fisher Scientific Inc., Waltham, MA, USA). The gene expression levels were normalized using β-actin (*Actb*) as the housekeeping gene. TaqMan hydrolysis primers and probe assays for the genes *KCNJ8* (ID: qHsaCIP0028892), *KCNJ11* (ID: qHsaCEP0055169), *ABCC8* (ID: qHsaCEP0049175), *ABCC9* (ID: qHsaCEP0049542), *TRPV1* (ID: qHsaCIP0033268), and *β-actin* (ID: qHsaCED0019162) were obtained from Bio-Rad (Milan, Italy).

*Western blot.* To extract proteins, cell pellets from control and treated samples were lysed with Pierce RIPA buffer (Thermo Scientific, Rockford, IL, U.S.A.) containing protease and phosphatase inhibitors (Thermo Scientific, Rockford, IL, U.S.A.). The lysates were homogenized and centrifuged at 14,000 rpm for 20 min at 4 °C. Protein concentrations were determined using the Bradford assay (Bio-Rad, Milan, Italy). Then, 25 µg of protein from each sample was denatured in a 5× Laemmli sample buffer and loaded onto pre-cast polyacrylamide gels (Bio-Rad, Milan, Italy) for Western blotting. Primary antibodies (Appendix A) were applied at a 1:1000 dilution, except for anti-β-actin, which was diluted to 1:4000. The membranes were incubated overnight with the primary antibodies, followed by incubation with a horseradish peroxidase-conjugated secondary antibody at a 1:5000 dilution. Proteins were detected using chemiluminescence (Clarity Western ECL substrate, Bio-Rad, Milan, Italy), and the signals were analyzed with the ChemiDoc System and Image Lab software 6.1 (Bio-Rad Laboratories Inc., Hercules, CA, USA). β-actin served as the reference for normalizing the densitometric values of each band (OD units).

*Data analysis and statistics.* Data concerning in vitro viability experiments were collected and analyzed using Excel software (Microsoft Office 2010) and GraphPad Prism 8. The statistical results are presented as mean ± SEM. One way ANOVA was used to test for variance between and within the groups. The number of replicates relative to each experimental dataset was reported in the “Results” paragraph and in the figures description as well as the number of biological samples. The Student’s *t*-test was used to evaluate the significance of differences between the means of two groups, and *p* values < 0.05 were considered to indicate statistical significance. Concentration-response curves were obtained using Graph Pad Prism 8. Graphs were built by setting non-linear regression, in a logarithmic form of drug concentrations, with normalization, so that the following was obtained: [Inhibition of Cell’s Proliferation(drug)/Inhibition of Cell’s Proliferation] = 1/(1 + ([Drug])/IC_50_)^n^. For patch clamp experiments on cells, the percentage of the reduction in currents was calculated as (I drug − I CTRL)/(I max − I CTRL) × (100) − 100, where I max was evaluated at the maximum voltage applied in the control (CTRL) condition; it was also calculated the percentage of reduction relative only to the control, which was: (I drug/I CTRL) × (100) − 100.

## 3. Results

### 3.1. In Vitro Cell Viability Experiments on DIPG-36 Cell Line Using Multi-Wells Crystal Violet Staining Assay

We first compared the effect of several ion channel modulators and some chemotherapeutic agents on the neuronal DIPG-36 cell line at different concentrations and at different incubation times to evaluate their capability to induce antiproliferative effects on these neuronal cells; these drugs were evaluated using multi-wells crystal violet staining test labeling DNA. On DIPG-36, doxorubicin reduced the percentage of cell survival by 27%, and 100% after 48 h and 72 h of incubation, respectively (Figure 1A–C).

Cisplatin (CISP) weakly reduced cell proliferation after an incubation time of 6 h. Instead, cell survival was reduced by 59% and 78% after 48 and 72 h of incubation time, respectively. After 6 h of incubation time, the KATP channel blockers glibenclamide (GLIB) and tolbutamide (TOLB) induced a reduction in cell proliferation by 15% and 24%, respectively, at 50 nM and 50 µM concentrations but without significance; glimepiride (GLIMP) (1 μM to 100 μM) was not effective. But repaglinide (REPA) (100 μM) at 48 and 72 h of incubation times reduced by 100% cell survival, demonstrating its effectiveness. The KATP opener DIAZO induced a not significant cell proliferation of +10% at 25 µM after 6 h of incubation time but a reducing cell proliferation of −10% at higher concentration of 250 µM. At this concentration, DIAZO reduced cell proliferation by 21% after 48 h of incubation. The 1 µM CAPS and 1 µM CAPSZ showed, respectively, +26% and −15% changes in cell proliferation after 6 h of incubation. A mild reduction in cell proliferation, respectively, of −15% and −17% was observed with higher CAPS concentration of 50 µM and 20 µM after 6 h and 48 h of incubation, respectively, but not observed at prolonged incubation times. After 48 h of incubation, GLIB reduced cell proliferation in a wide range of concentrations of 100 nM–100 µM but not significantly by ANOVA test, while GLIMP markedly reduced cell proliferation at 100 µM concentration (Figure 1B). All ion channel modulators induced cell proliferation after 72 h of incubation time at all concentrations evaluated with the exception of 100 μM REPA (Figure 1C).

In a clonogenic assay performed using the DIPG-36 cell line, REPA 100 μM reduced the number of cell colonies both after 72 h and 96 h of incubation, with a surviving fraction of 54 ± 0.05% after 72 h and of 49 ± 0.06% after 96 h vs. controls. The number of colonies were 50 in the controls and 27 with REPA after 72 h, and 345 in the controls and 170 with REPA after 96 h (Figure 2A, B). A low CAPS concentration failed to affect the number of colonies with 172 formed colonies after 72 h vs. 164 in the controls, and with a surviving fraction of 104.8 ± 0.05% (Figure 2C).

The antiproliferative effect of the most effective substances, REPA (0.1 μM to 500 μM), and GLIB (0.1 μM to 500 μM), was further evaluated on neuronal DIPG-36 cell lines at two different incubation times by concentration-response curves analysis. REPA gave concentration-dependent responses, and we obtained a normalized percentage of inhibition of cell proliferation equal to 0 with 100 μM REPA after 48 h of incubation using multi-wells crystal violet assay labeling of the DNA using instead the CCK-8 assay to evaluate the drug’s action on mitochondrial dehydrogenase activity, REPA was not shown to be as effective as it is with the multi-well crystal violet assay (Figure 3A,B).

Furthermore, after an incubation time of 72 h we observed a normalized percentage of the inhibition of cell proliferation equal to 0 already using 100 μM REPA with multi-well crystal violet assay; with CCK-8; however, the most effective concentration is 200 μM, obtaining a normalized percentage of inhibition of cell proliferation of 9.32%. GLIB provided time-dependent incubation results with the CCK-8 test, with a normalized percentage of cell proliferation inhibition of 36.95%, using 200 μM GLIB after 48 h of incubation and achieving 0 using the same concentration after an incubation time of 72 h (Figure 3B). No GLIB data was provided in CV assay. GLIB in the CCK-8 assay shows overlapping concentration-response curves with REPA in the CCK-8 assay. The calculated IC_50_ (M) supports the conclusion (Table 1).

### 3.2. Characterization of Cation Channel Currents of DIPG-36 Cells

Three different morphological cell types were found in the sampled DIPG-36 populations (number of cells = 60 cells), conventionally defined as multipolar (Figure 4A), bipolar (Figure 4B), and starry (Figure 4C) based on their aspect. Bipolar cells were the most abundant in the sampled population (Figure 4D). Using physiological K^+^ ion concentrations in the bath and in the pipette solutions, a hyperbolic current–voltage relationship is recorded in whole-cell patches. The resting potential (Vm) of these cells was a −10.2 ± 8 mV characteristic of tumor cells rather than normal neurons. The membrane capacitance was 42.02 ± 12 pF (number of cells = 60), and the mean currents of large intensity were inhibited by BaCl_2_ (10 mM) and TEA (5 mM) with depolarization (Figure 4E). We also found cells (number of cells = 4) with inward rectifications of the currents at negative membrane potentials and resting potentials close to the equilibrium potentials for K^+^ ions (Figure 4F) resembling normal neurons.

The KATP opener DIAZO (250 μM) applied on the external side of the cells enhanced the currents (Figure 5A,B), respectively, of +20% at +30 mV (Vm) and +27% at −30 mV (Vm), that were reduced, respectively, of −28% at +30 mV (Vm) and of −34% at −30 mV (Vm) by −10 μM GLIB, a selective KATP inhibitor (Figure 5B). A significant reduction was seen also at different negative membrane potentials suggesting a contribution of KATP channel currents to the control currents (number of cells = 10). The incubation of the cells with 10 mM BaCl_2_ and 5 mM TEA caused a marked reduction in the control currents especially at positive potentials (Figure 5B) and a minor reduction at negative membrane potentials.

On seven cells, the application of 10 μM CAPS, a well-known TRPV1 channel agonist, failed to enhance the outward currents suggesting that the channel is already active. 1 μM CAPSZ, a selective TRPV1 channel blocker, reduced the outward currents of −16.43 ± 2.1% at +80 mV (Vm) (Figure 5C,D). The highest reduction was obtained at +180 mV (Vm), which is the voltage at which most of the TRPV1 channels are activated and was of −31.9 ± 2.6% supporting the idea that the TRPV1 channel is functionally active in these cells. At positive membrane potentials, the currents were further reduced by 10 μM ruthenium red (RR), a non-selective TRP channel blocker (number of cells = 7). The application to the cells of 10 mM BaCl_2_ and 5 mM TEA fully reduced the control currents (Figure 5D).

Therefore, the control currents of the DIPG-36 cells recorded in the physiological condition were enhanced by diazoxide and reduced by the KATP channel inhibitor GLIB 10 μM supporting the presence of functional KATP channels. The currents were also reduced by the TRPV1 antagonist CAPSZ 1 μM at positive membrane potentials. At these concentrations of inhibitors, the TRPV1 and KATP channels are expected to be fully inhibited according to the IC_50_ values of about 500 nM for CAPSZ and 1 nM–0.1 μM for GLIB, respectively, on ion channels expressed in different cells or in artificial membrane [36,37]. The non-selective inhibitors of the potassium channels BaCl_2_ and TEA were the most effective compounds in reducing the control currents in DIPG-36 cells suggesting that other cation channels are active in this cell type. It should be of note that the observed partial reduction in the control currents in DIPG-36 cells by the TRPV1 channel inhibitors is in line with the mild effects on cell proliferation after 6 h of incubation time in the same cells.

### 3.3. “In Vitro” Cell Viability Experiments on DIPG-50 Cell Line Using Multi-Wells Crystal Violet Staining Assay

The drug effects were evaluated at different concentrations and at three different incubation times (6 h, 48 h, and 72 h) in a multi-well crystal violet assay to understand their capability to induce antiproliferative effects on these neuronal cells. No clonogenic assay could have been performed in these cells because of the much longer incubation time of 96 h needed for this experiment and because of the elevated number of not adherent cells in these cell cultures. In regard to chemotherapeutic agents, we used cisplatin (0.1 mM) (CISP), which reduced the percentage of cell survival by 9% after 6 h and has not shown a cytotoxic effect after both 48 h and 72 h of incubation. Instead, 10 μM DOXO, did not demonstrate cytotoxicity after 6 h of incubation but was more cytotoxic as the incubation time increases: in fact, after 48 h of incubation it reduced the percentage of cell survival by 70%, and after 72 h by 100%. Among the potassium channel modulators, we used DIAZO at different concentrations (250 nM to 250 μM) and GLIB (100 nM to 100 μM) and DIAZO after 6 h and 48 h of incubation induced a marked but a not significant cell proliferation (Figure 6A,B).

GLIB has not demonstrated antiproliferative activity after 6 h of incubation; instead, after an incubation time of 48 h, 100 μM GLIB reduced cell survival percentage by 46%. At this concentration, it was also effective after 72 h of incubation. Furthermore, it has also been evaluated that tolbutamide only had a marked but not significant proliferative effect after all the incubation times in this cell type (Figure 6A–C). Instead, repaglinide (100 μM–200 μM) (REPA) reduced cell proliferation at all the incubation times. After 6 h of incubation time, cell survival was reduced by 30% using 100 μM REPA, and by 39% using 200 μM REPA; the best result was the one available using REPA 200 μM after an incubation time of 72 h, reducing cell survival percentage by 89%. Furthermore, GLIMP (100 μM–200 μM) failed to reduce cell proliferation after 6 h and 48 h of incubation. Instead at 72 h of incubation time, 200 μM GLIMP reduced cell survival by 37%. Capsaicin (50 μM to 100 μM) (CAPS) was also evaluated, and after an incubation time of 72 h, at the highest concentration (100 μM), reduced cell survival by 35%. Capsazepine (1 μM to 10 μM) (CAPSZ) did not affect cell viability at all the incubation times (Figure 6A–C).

In a clonogenic assay performed in the DIPG-50 cells, REPA 100μM reduced the number of cell colonies with a surviving fraction of 57.46 ± 0.05% vs. controls after 96 h of incubation. The number of colonies were 142 in controls and 81 with REPA after 96 h (Figure 6A). A low CAPS concentration of 1µM also reduced colonies, with 106 formed colonies vs. 142 in controls, with a surviving fraction of 74.48 ± 0.05% after 96 h (Figure 6D).

### 3.4. Apoptosis Assays

A concentration-dependent apoptosis was confirmed by flow cytometry analysis of DIPG cells in the presence of REPA 1 and 100 μM after 48 h of incubation time (Figure 7A,B). How cell diameter changed as repaglinide was added to cell cultures was also investigated, distinguishing cell populations into two diameter size groups: one cell population with a diameter from 6 μm to 11 μm and the other cell population with a diameter from 11 μm to 24 μm. After an incubation time of 48 h, there was a noticeable decrease in the number of cell population with a diameter between 11 μm and 24 μm (Appendix A) when 200 μM repaglinide was used; instead, cell population with a diameter from 6 μm to 11 μm was increased, suggesting of an apoptotic cell death. After 72 h of incubation, using 200 μM repaglinide, both cell populations were decreased, demonstrating a strong cytotoxicity and cell death. Also, cell volume showed a significant decrease when 200 μM repaglinide was used, and this condition confirms the marked cell death caused by repaglinide (Appendix A).

Through concentration-response curves, we investigated the effect of the most effective drugs, REPA and GLIB, on the neuronal DIPG-50 cell line at different concentrations (0.1 μM to 500 μM), and at two different incubation times, to evaluate their capability to induce antiproliferative effects on these neuronal cells. Using 500 μM REPA, a normalized percentage of inhibition of cell proliferation equal to 0 was obtained after 48 h of incubation using both multi-wells crystal violet and CCK-8 assays. REPA was also effective at lower concentrations using CCK-8 assay, bringing the normalized percentage of inhibition of cell proliferation to 0 (Figure 8A,B). In addition, after an incubation time of 72 h, a normalized percentage of inhibition of cell proliferation of 0 already using 100 μM REPA was achieved using both multi-wells crystal violet and CCK-8 assay. GLIB also had concentration-dependent behavior, with a normalized percentage of inhibition of cell proliferation of 0 using 500 μM GLIB after an incubation time of 48 h, with both assays. After an incubation time of 72 h, a normalized percentage of inhibition of cell proliferation of 0 was obtained with multi-well crystal violet assay and CCK-8 using 200 μM and 500 μM GLIB (Figure 8A–B. REPA and GLIB were, therefore, more potent in DIPG-50 vs. DIPG-36 and the IC_50_ values of the drugs confirmed these conclusions (Table 1 and Table 2).

In mouse bone marrow cells REPA (0.1 μM to 500 μM) and GLIB (0.1 μM to 500 μM) at increasing concentrations did not significantly affect cell proliferation after 72 h of incubation times in multi-wells crystal violet (CV) assay (Appendix A).

### 3.5. Real Time Polymerase Chain Reaction (RT-PCR) Data on Drug Target Ion Channel Gene Subunits in DIPG-36 and DIPG-50 Cells, and Response of Control Currents to Repaglinde in DIPG-50 Cells

The expression of KATP and TRPV1 channel subunits in the DIPG-36 and DIPG-50 cell lines, using RT-PCR experiments were conducted. Beta-actin was used as the housekeeping gene. As a result, the *ABCC8* gene was found to be expressed in both cell lines, with higher expression in DIPG-50; *ABCC8* encodes for the high affinity sulfonylurea receptor 1 (SUR1). Instead, the *ABCC9* gene is expressed in both cell lines, but with a greater extent in DIPG-36; *ABCC9* gene encodes for the lower affinity sulfonylurea receptor 2 (SUR2). *KCNJ11* was found to be expressed in both cell lines, although in lower quantities than the other studied genes; this gene encodes for Kir6.2 subunit. *KCNJ8* is highly and equally expressed in both cell lines; this gene encodes for the Kir6.1 subunit. Based on these data, we can therefore confirm the presence of KATP encoding subunits within both DIPG-36 and DIPG-50 cell lines (Figure 9A). The gene expression data are in line with the higher potency of REPA and GLIB as the antiproliferative drugs seen in the DIPG-50 vs. the DIPG-36. *TRPV1* gene is expressed in both cell lines with a stronger expression in DIPG-50 cells, which further confirms the presence of this channel in both cells (Figure 9A).

Using a physiological K^+^ ion concentration in the bath and in the pipette solutions, a hyperbolic current–voltage relationship was recorded in whole-cell patches in the presence of 250 μM DIAZO (Figure 9B). The resting potential (Vm) of these cells was −8.2 ± 2 mV, the membrane capacitance was 41.02 ± 9 pF (cell number = 10), and the mean currents of large intensity, which were inhibited by 10 mM BaCl_2_ and 5 mM TEA with depolarization, were recorded as observed in the DIPG-36 cells. The selective KATP antagonist 100 μM REPA at +60 mV (Vm) reduced the currents potentiated by 250 μM DIAZO (178.50 ± 16.6 pA), with a percentage of −27.62%; instead at −80 mV (Vm) a reduction of the control current (−60.26 ± 15.2 pA) of 40.03% was seen (Figure 9B). The incubation of cells with 10 mM BaCl_2_ and 5 mM TEA caused a reduction in the control currents, which is closer to positive potentials (Figure 9B). It should be of note that these cells are composed of adherent and not adherent cell populations that are not available for patch clamp analysis, reducing the number of sampled cells.

### 3.6. Western Blot Results on Target Protein Content in DIPG-36 and DIPG-50 Cells

Western blot was performed to evaluate the target proteins content in DIPG-36 and DIPG-50 cell lines; beta-actin was used as the normalization standard in these experiments. In the first Western blot, the H3K27ac protein content, i.e., the acetylation of the lysine at position 27 of histone H3, was evaluated. The H3K27ac protein increased in a concentration-dependent way in both cell lines in the presence of 1 μM and 100 μM REPA (Figure 10).

We further investigated the specificity action of REPA on H3K27ac vs. the Tri-Methyl H3(Lys27), which is also a key epigenetic element regulating histone, but no significant changes were observed with different concentrations of REPA in DIPG cells (Appendix A). (The original images of uncropped Western blot and molecular weight markers are shown in the Appendix A).

This effect can be associated with the inhibition of histone deacetylase or other enzyme downregulating mechanisms by REPA. We, therefore, further tested this hypothesis in DIPG cells treated with different REPA concentrations after 48 h of incubation, but we failed to show an enhancement of SUR2 protein in Western blot experiments, rather a tendency of SUR protein content reduction was observed, which was not significant (Appendix A) (The original images of uncropped Western blot and molecular weight markers are shown in the Appendix A).

Caspase-3 is an inactive pro-enzyme, which in its inactive form has no role in apoptosis; its active form cleaved-caspase 3 is formed by a proteolytic cleavage during the initiation of apoptosis through various signaling pathways. Thus, this activated enzymatic form plays an important role in apoptosis. In DIPG-36 and DIPG-50 cells, a significantly elevated content of cleaved casp-3 was found in the presence of 1 μM and 100 μM REPA after 48 h of incubation time. The activated casp-3 content agrees with the observed antiproliferative action of REPA in DIPG-36 and DIPG-50 cells (Figure 11).

We finally investigated the possible effect of REPA on some intracellular signaling in DIPG as a secondary aim. Dysregulation of mTOR exists in several diseases, including cancer, because its overactivation can promote tumor growth and survival. Phosphorylation of mTOR at Ser2448 is associated with its activation, so its kinase activity increases, leading to the phosphorylation of downstream targets involved in cell growth, proliferation, and metabolism. Dysregulation of mTOR signaling, including phosphorylation of Ser2448, is common in cancer. Hyperactivation of mTOR promotes tumor growth, survival, and metastasis by stimulating cell proliferation, angiogenesis, and resistance to apoptosis. In our experiments, the total mTOR content in DIPG-36 cells decreases after treatment with 1 μM REPA, and a slight decrease is also achieved with treatment with 100 μM REPA. Treatment with 1 μM and 100 μM REPA induces a marked reduction in phospho-mTOR, which may show that this is an additional mechanism of action of REPA, which contributes to the antiproliferative effect (Figure 12). Instead, by treating DIPG-50 cells with 1 μM and 100 μM REPA, the total mTOR and phospho-mTOR are increased. Then, for the DIPG-50 cell line, this cannot be identified as a mechanism of action of REPA antiproliferative effects (Figure 12). 

Dysregulation of AKT signaling, including phosphorylation at Ser473, is commonly seen in cancer. In the DIPG-36 cell line, REPA has a concentration-dependent effect in decreasing the level of inactivated total AKT but not of the activated form phospho-AKT (Ser473) (The original images of uncropped Western blot and molecular weight markers are shown in the Appendix A).

ERK p42 (Extracellular Signal-Regulated Kinase) and ERK p44, respectively, ERK 1 and ERK 2, were upregulated by REPA (The original images of uncropped Western blot and molecular weight markers are shown in the Appendix A).

So, based on these results, we do not consider this route responsible for the anti-proliferative mechanism of REPA. No changes in the expression were observed in other receptors such as PDGR and VEGFR, and in the Beclin and LC3 contents with REPA and were not further investigated (The original images of uncropped Western blot and molecular weight markers are shown in the Appendix A).

## 4. Discussion

In this study, we investigated the effects of ion channel modulators on the human DIPG-36 and DIPG-50 cell lines, using “in vitro” cell survival experiments, patch clamp analysis, RT-PCR gene expression and Western blot investigations.

First, we made a screening test “in vitro,” to understand which drugs had the most effective antiproliferative effect on these two cell lines; among those substances we evaluated that the most interesting effects were those of repaglinide and glibenclamide, which are well-known inhibitors of KATP channels available for clinical use.

We, therefore, evaluated their effects on ion channels currents through patch clamp experiments. Of note, this is the first report investigating the ion channel currents in native DIPG-36 and DIPG-50 cells and their response to KATP and TRPV1 channel modulators. All sampled cells were sensitive to KATP channel modulators, the control currents potentiated by diazoxide were reduced by these drugs; for instance, 10 µM glibenclamide was evaluated on the DIPG-36 with a maximal current reduction of about 34% of the whole cell current in intact cells. Also, 100 µM REPA reduced the diazoxide potentiated currents by 40.03% in DIPG-50 cells. The reduction of the channel currents by the KATP inhibitors is in line with their antiproliferative effects observed after 6 h of incubation in either DIPG-36 and DIPG-50 cells. In another groups of cells, 1 μM capsazepine reduced the membrane currents by 31% that were markedly reduced by ruthenium red, suggesting that the presence of active TRP channels also contributed to these currents. The observed partial reduction in the control currents with the TRPV1 inhibitor is in line with a mild reduction in cell proliferation in DIPG-36 cells after 6 h of incubation.

KATP channels subunits are emerging drug targets in some cancers [38]. We showed that the marked cell survival decrease obtained with repaglinide after incubation periods of 48 h and 72 h is better correlated with the concentration-related enhanced content of the H3K27ac protein in DIPG cells. This can result in the inhibition of the histone deacetylase. SUR2 is functionally coupled to the HDCA gene in cardiomyocyte [39], and it may also be coupled in neurons. The inhibition of the histone deacetylase by histone deacetylase inhibitors (HDAC-I) is associated with antiproliferative effects in cancers so that some HDAC-I drugs are under investigation in DIPG [40]. This target protein is a marker of genes involved as epigenetic regulators in cancers, neurological diseases, and diabetic kidney disease, and H3K27ac is associated with genes transcription involved in the pyroptotic cell death [41,42,43,44]. In our DIPG cells, the enhancement of the H3K27ac protein is associated with upregulation of cleaved caspase 3 and apoptosis supported by the reduced cell diameter following repaglinide treatment. The H3K27ac has a role in cancers [45]. The enhancement of the H3K27ac with apoptosis has been recently reported in sarcoma [46].

In addition, the marked antiproliferative effects of KATP channel inhibitors observed at longer incubation times in DIPG cells can be due to the interaction with additional non-surface membrane targets. For instance, mitochondrial KATP channel subunits are expressed in the inner membrane and are coupled to the dehydrogenase activity [47]. The mitochondria are considered relevant organelles in cancers and in DIPG, so the Human Mitochondrial Caseinolytic Serine Protease activation is a proposed target for DIPG treatment [48]. The inhibition of dehydrogenase activity can explain the higher potency of repaglinide found in the DIPG-50 cells vs. DIPG-36. In addition, the higher sensitivity of the DIPG-50 cells to KATP inhibitors vs. the DIPG-36 can be related to the higher expression levels of the SUR1 subunit encoded by the *ABCC8* gene carrying the high affinity site for sulfonylureas than the SUR2 subunit encoded by the *ABCC9* gene carrying the low affinity site for the drugs that indeed is more expressed in the DIPG-36.

Regarding the secondary aims of our study, a significant downregulation of the phospho-mTOR signaling was found in DIPG cells particularly in DIPG-36 after 48 h of incubation time by Western blot with repaglinide.

## 5. Conclusions

In conclusion, in this study, we have evaluated the antiproliferative effect of KATP channels inhibitors such as glibenclamide and repaglinide and TRPV1 channels modulators on DIPG-36 and DIPG-50 cell lines. Moreover, we have also characterized channel currents never reported before, proving the presence of two ion channel families (KATP and TRPV1 channels) in these cell populations also confirmed by the elevated gene expression of the relative KATP and TRPV1 channel gene-subunits.

Metformin has been recently proposed in DIPG in combination with paxalisib and enzastaurin to overcome the hyperglycemia associated with the use of Pi3-kinase inhibitors paxalisib [49], but metformin, as pointed out in the commentary published on J Clin Investigation of Theophilos Tzaridis and Robert J. Wechsler-Reya [50], is used at very high doses and is not anti-proliferative “per se”. In our study we propose a combination of paxalisib and enzastaurin with repaglinide or glibenclamide that show hypoglycemic effects and direct antiproliferative effects in DIPG cells.

Also, mild antiproliferative effects were observed with the TRPV1 antagonist capsazepine after 6 h of incubation in DIPG-36 cells and with the TRPV1 agonist capsaicin after 72 of incubation in DIPG-50 cells suggesting a possible role of this receptor in regulating cell proliferation in these cells. New strategies are emerging [51].

## Figures and Tables

**Figure 1 cancers-17-00358-f001:**
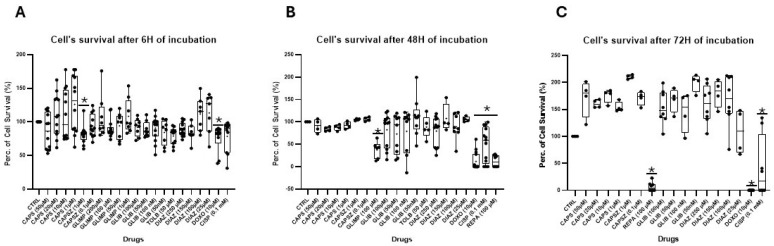
Percentage changes in cell survival in multi-wells crystal violet (CV) assay in DIPG-36 cells. (**A**) Capsaicin (CAPS) (10 μM to 50 μM), capsazepine (CAPSZ) (0.1 μM to 1 μM), glimepiride (GLIMP) (1 μM to 200 μM), glibenclamide (GLIB) (50 nM to 100 μM), tolbutamide (TOLB) (50 μM), diazoxide (DIAZO) (25 μM to 250 μM), doxorubicin (DOXO) (10 μM), and cisplatin (CISP) (0.1 mM) data after 6 h of incubation. REPA was not evaluated. (**B**) Capsaicin (CAPS) (1 μM to 50 μM), capsazepine (CAPSZ) (0.1 μM to 1 μM), repaglinide (REPA) (100 μM), glimepiride (GLIMP) (1 μM to 100 μM), glibenclamide (GLIB) (50 nM to 100 μM), tolbutamide (TOLB) (50 μM), diazoxide (DIAZO) (25 μM to 250 μM), doxorubicin (DOXO) (10 μM) and cisplatin (CISP) (0.1 mM) data after 48 h of incubation. (**C**) Capsaicin (CAPS) (1 μM to 50 μM), capsazepine (CAPSZ) (0.1 μM to 1 μM), repaglinide (REPA) (100 μM), glibenclamide (GLIB) (50 nM to 100 μM), diazoxide (DIAZO) (25 μM to 250 μM), doxorubicin (DOXO) (10 μM) and cisplatin (CISP) (0.1 mM) data after 72 h of incubation. The data are the means of at least three samples and six replicates. Data are significantly different within and between groups with a one-way ANOVA test with F values > 2 (*).

**Figure 2 cancers-17-00358-f002:**
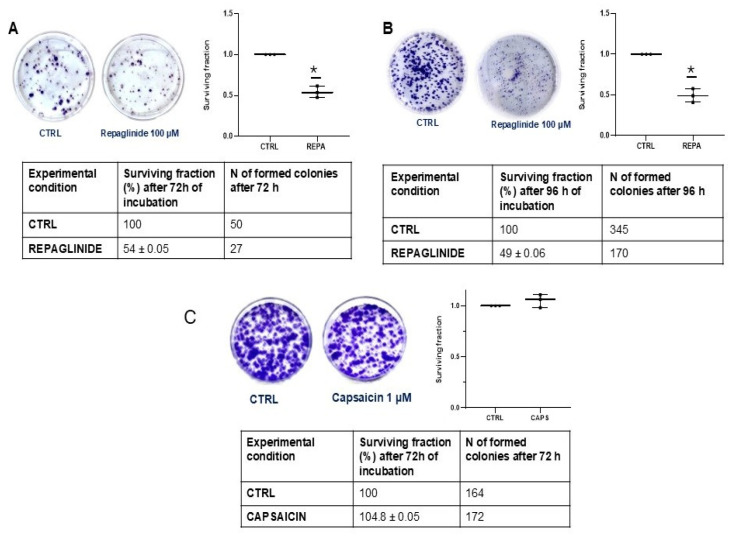
Clone formation with repaglinide and capsaicin in DIPG-36. Effect of 100 µM repaglinide (REPA) on DIPG-36 cell clone formation evaluated after (**A**) 72 and (**B**) 96 h of incubation times using the clonogenic assay and the effect of low concentration of (**C**) capsaicin (CAPS) 1 μM after 72 h of incubation. The data are from at least three samples.* Data significantly different vs controls by Student *t* test, for *p* < 0.05.

**Figure 3 cancers-17-00358-f003:**
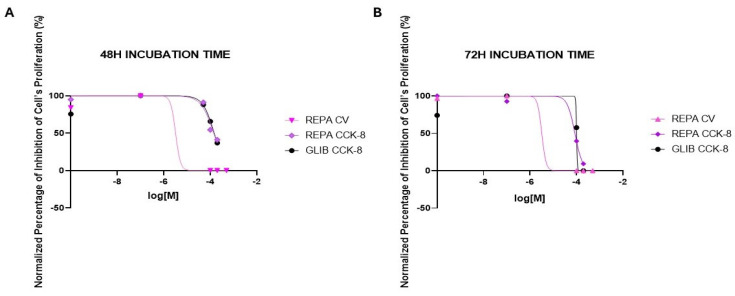
Concentrationresponse curves of the normalized percentage (%) inhibition of cell proliferation vs. increasing concentrations of glibenclamide (GLIB) and repaglinide (REPA) on DIPG-36 cells obtained by multi-wells crystal violet (CV) and CCK-8 assays. (**A**) After 48 h of incubation, and (**B**) after 72 h of incubation.

**Figure 4 cancers-17-00358-f004:**
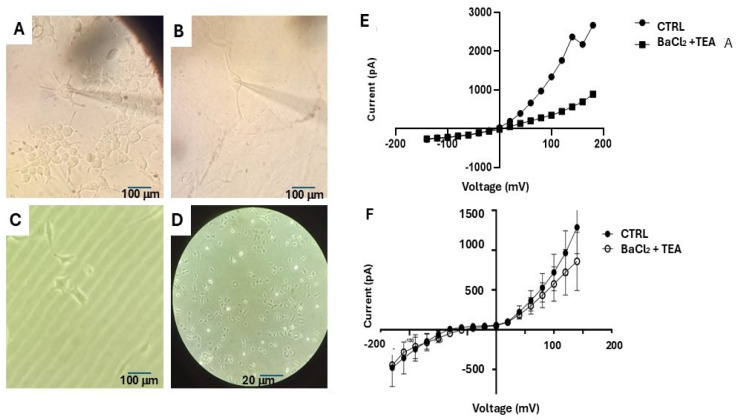
Cation-channel currents in DIPG-36 cells in whole cell patch clamp. (**A**) Multipolar cells in the human DIPG-36 cell culture, 20×. (**B**) Bipolar cell in the DIPG-36 population, 20×. (**C**) Starry cell, 20×. (**D**) DIPG-36 cells, 5×. (**E**) Mean current–voltage relationships (Number of cells = 60). The mean resting potential of these cells was −10.2 ± 8 mV (Vm) and was −2 mV ± 3 mV (Vm) with BaCl_2_ and TEA. (**F**) Mean current–voltage relationships (Number of cells = 4). The mean resting potential of these cells was −90.2 ± 8 mV (Vm) and was −60 mV ± 3 mV (Vm) with BaCl_2_ and TEA resembling normal neurons.

**Figure 5 cancers-17-00358-f005:**
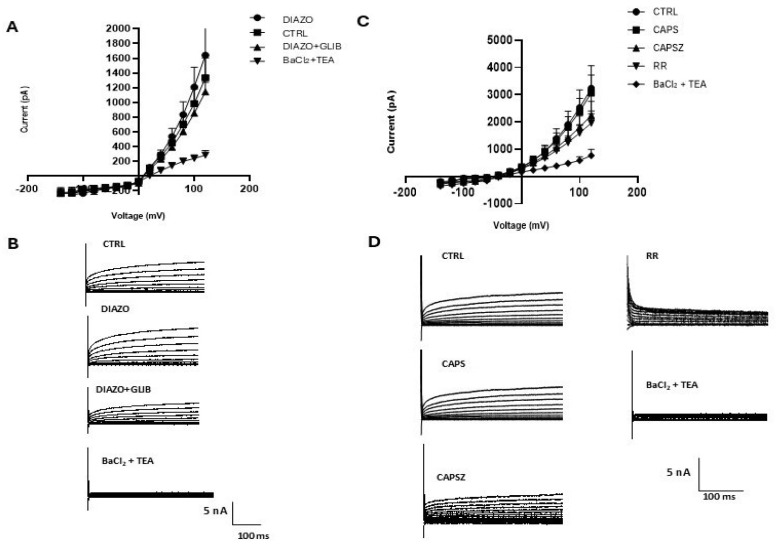
Effects of KATP channel and TRPV1 modulators on cation channel currents in DIPG-36 cells. Left: Effects of KATP modulators on (**A**) current–voltage relationship and (**B**) whole cell currents recorded in physiological condition using the patch clamp technique on 10 cells. The KATP opener DIAZO (250 μM) was applied on the external side of the cells, and it was evaluated after DIAZO+GLIB at a 10 μM concentration. Finally, cells were incubated with 10 mM BaCl_2_ and 5 mM TEA. Right: Effects of TRPV1 modulators on DIPG-36 whole cell currents recorded in physiological condition on seven cells. (**C**) Current–voltage relationships, and (**D**) percentage changes in current reduction following the application of TRPV1 modulators. Sample current traces representing the effects of the TRPV1 channel agonist 10μM CAPS that was applied on the external side of the cells, followed by 1 μM CAPSZ, a selective TRPV1 channel blocker; in addition, 10 μM ruthenium red (RR), a non-selective TRP channel blocker, was applied. Finally, cells were incubated with 10 mM BaCl_2_ and 5 mM TEA to reduce the residual K^+^ channel currents.

**Figure 6 cancers-17-00358-f006:**
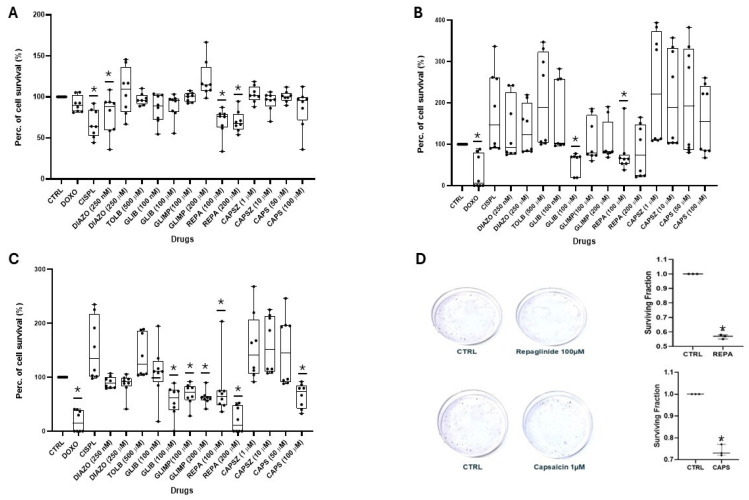
Percentage changes in cell survival of DIPG-50 cells in multi-well crystal violet (CV) and clone formation in clonogenic assays. (**A**) CAPS (50 μM to 100 μM), CAPSZ (1 μM to 10 μM), REPA (100 μM to 200 μM), GLIMP (100 μM to 200 μM), GLIB (100 nM to 100 μM), 500 nM TOLB, DIAZO (250 nM to 250 μM), 10 μM DOXO and 0.1 mM CISP data after 6 h of incubation. (**B**) CAPS) (50 μM to 100 μM), CAPSZ (1 μM to 10 μM), REPA (100 μM to 200 μM), GLIMP (100 μM to 200 μM), GLIB (100 nM to 100 μM), 500 μM TOLB, DIAZO (250 nM to 250 μM), 10 μM DOXO and 0.1 mM CISP data after 48 h of incubation. (**C**) CAPS (50 μM to 100 μM), CAPSZ (1 μM to 10 μM), REPA (100 μM to 200 μM), GLIMP (100 μM to 200 μM), GLIB (100 nM to 100 μM), 500 μM TOLB, DIAZO (250 nM to 250 μM), 10 μM DOXO and 0.1 mM CISP data after 72 h of incubation. (**D**) Effects of 100 µM repaglinide (REPA) and 1 µM capsaicin on DIPG-50 cell clone formation evaluated after 96 h of incubation time using the clonogenic assay in the presence of Matrigel. The data are the mean ± ES of at least three samples and six replicates. Data are significantly different within and between groups with one-way ANOVA test with F values > 2 (*).

**Figure 7 cancers-17-00358-f007:**
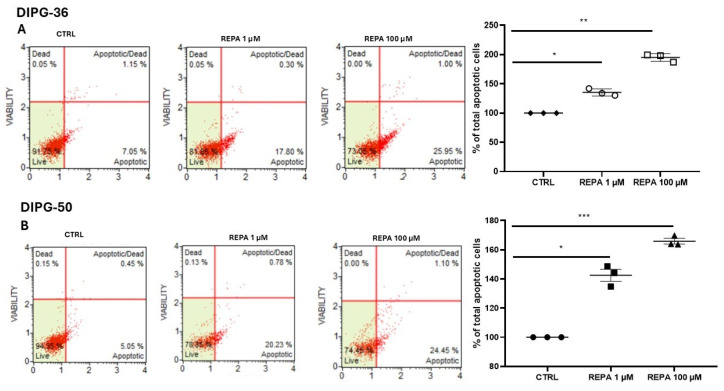
DIPG-36 (**A**) and DIPG-50 (**B**) cells were treated with REPA 1 and 100 μM for 48 h, and apoptosis was detected by flow cytometry using the CellEvent™ Caspase-3/7 Green Assay Kit (Thermofisher, C10427). Data are presented as the mean ± standard deviation of three independent repeats, * *p* < 0.05, ** *p* < 0.005, *** *p* < 0.001 vs. controls.

**Figure 8 cancers-17-00358-f008:**
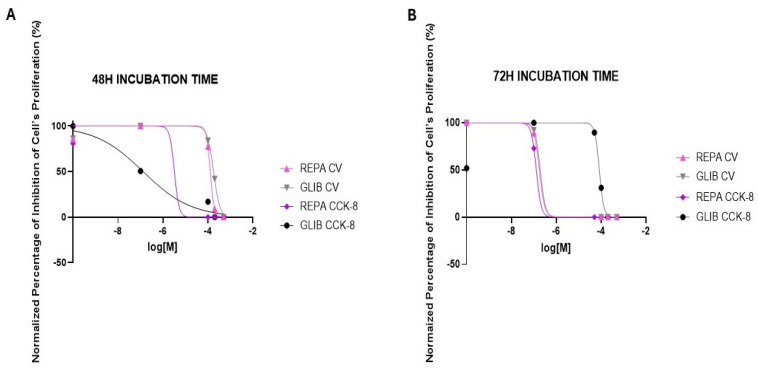
Concentration-response curves of the normalized percentage (%) inhibition of cell proliferation vs. increasing concentrations of glibenclamide (GLIB) and repaglinide (REPA) on DIPG-50 cells obtained by multi-wells crystal violet (CV) and CCK-8 assays (**A**) after 48 h of incubation and (**B**) after 72 h of incubation.

**Figure 9 cancers-17-00358-f009:**
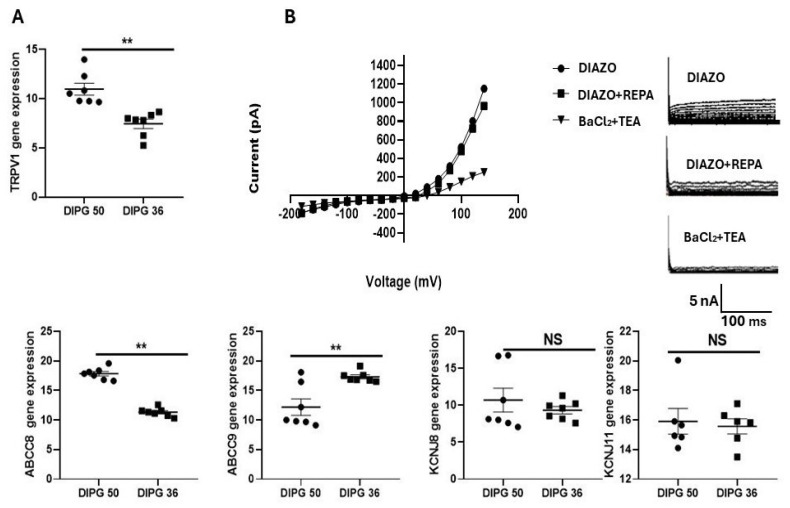
Gene-expression of ion channel genes in DIPG cells in RT-PCR experiments and the response of potassium channel currents to repaglinide (REPA) in whole cell patch clamp in the presence of diazoxide in DIPG-50 cells. (**A**) Gene expression data are reported as the mean ± SEM of six independent sample experiments. Data was significantly different by Student’s *t*-test: ** *p* < 0.05 vs. the other group. (**B**) Current–voltage relationships and whole cells mean currents recorded in physiological conditions and using patch clamp technique, on DIPG-50 (number of cells = 10). The effect of the KATP antagonist REPA (100 μM) was evaluated in the presence of 250 μM DIAZO, and in the end, cells were incubated with 10 mM BaCl_2_ and 5 mM TEA. NS, data not significantly different.

**Figure 10 cancers-17-00358-f010:**
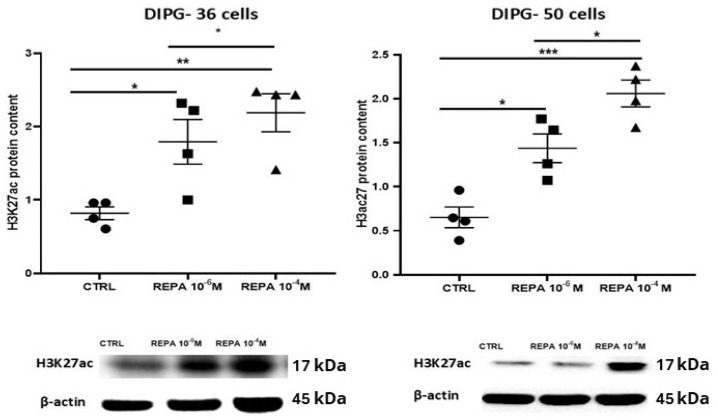
Effects of repaglinide after 48 h of incubation on H3K27ac content in DIPG cells. Above: acetyl histone H3K27ac content in DIPG-36 after treatment with REPA at different concentrations. It is reported the relative acetyl-histone H3 (Lys27) content normalized to β-actin after 48 h of treatment. Below: there are Western blot images showing acetyl histone H3K27ac protein after 48 h of treatment. β-actin was used as a housekeeping protein. Data are from the mean ± SEM of three independent experiments, * *p* < 0.05, ** *p* < 0.02, *** *p* < 0.01, vs. the control group by Student’s *t*-test. **Above**: Acetyl histone H3K27ac content in DIPG-50 after treatment with REPA at different concentrations and relative acetyl-histone H3 (Lys27) content normalized to β-actin content after 48 h of treatment. (The original images of uncropped Western blot and molecular weight markers are shown in the Appendix A). **Below**: there are Western blot images, which show acetyl histone H3K27ac protein after 48 h of treatment. β-actin was used as a housekeeping protein. Data are reported as the mean ± SEM of three independent experiments, * *p* < 0.05, ** *p*<0.02, *** *p* < 0.01, vs. the control group by Student’s *t*-test.

**Figure 11 cancers-17-00358-f011:**
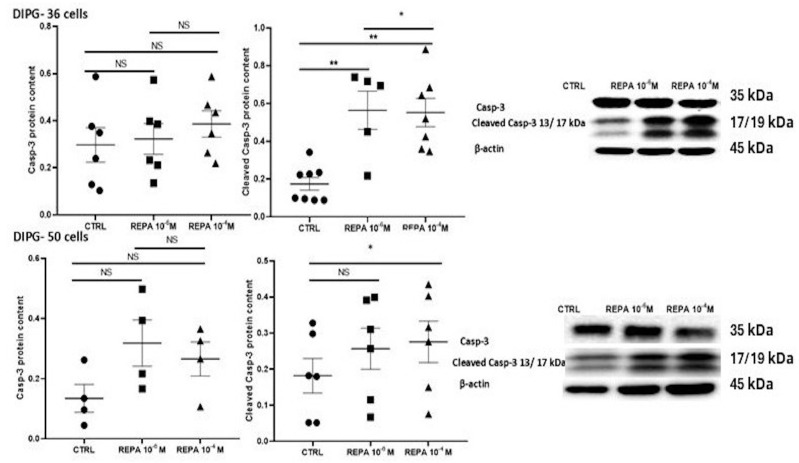
Effects of repaglinide after 48 h of incubation on cleaved caspase-3 in DIPG cells. Above: Caspase-3 (Casp-3) and cleaved caspase-3 (Cleaved Casp-3) in DIPG-36 after treatment with REPA at different concentrations. Relative caspase-3 and cleaved caspase-3 content normalized to β-actin after 48 h of treatment. Western blot images show caspase-3 and cleaved caspase-3 protein content after 48 h of treatment. β-actin was used as a housekeeping protein. Data are reported as the mean ± SEM of three independent experiments. Data significantly different for * *p* < 0.05, ** *p* < 0.01, vs. the control group by Student’s *t*-test. Below: Casp-3 and Cleaved Casp-3 in DIPG-50 after treatment with REPA at different concentrations. Relative Casp-3 and Cleaved Casp-3 content normalized to β-actin after 48 h of treatment. Western blot images show Casp-3 and Cleaved Casp-3 protein content after 48 h of treatment. β-actin was used as a housekeeping protein. Data are reported as the mean ± SEM of three independent experiments. Data were significantly different for * * p*< 0.05, ** *p* < 0.01, vs. the control group by Student’s *t*-test. NS, data not significantly different. (The original images of uncropped Western blot and molecular weight markers are shown in the Appendix A).

**Figure 12 cancers-17-00358-f012:**
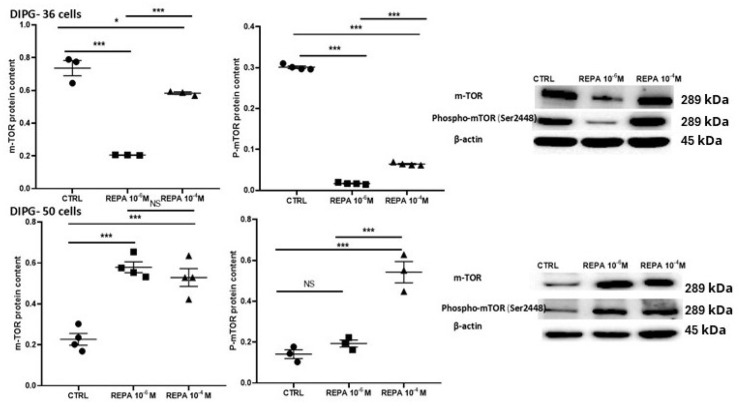
Effects of repaglinide on mTOR and phospho–mTOR (Ser2448) in DIPG cells. Above: the content of mTOR and phospho–mTOR (Ser2448) (P-mTOR) in DIPG-36 after treatment with REPA at different concentrations. Relative of mTOR and phospho–mTOR content normalized to β-actin after 48 h of treatment. Western blot images show mTOR and phospho-mTOR protein content after 48 h of treatment. β-actin was used as a housekeeping protein. Data are reported as the mean ± SEM of three independent experiments. Data are significantly different for * *p* < 0.05, *** *p* < 0.01, vs. the control group by Student’s *t*-test. Below: the content of mTOR and phospho–mTOR (Ser2448) in DIPG-50 after treatment with REPA at different concentrations. Relative of mTOR and phospho–mTOR content normalized to β-actin after 48 h of treatment. Western blot images show mTOR and phospho-mTOR protein content after 48 h of treatment. β-actin was used as a housekeeping protein. Data are reported as the mean ± SEM of three independent experiments. Data are significantly different for * *p* < 0.05, *** *p* < 0.01, vs. the control group by Student’s *t*-test. NS, data not significantly different. (The original images of uncropped Western blot and molecular weight markers are shown in the Appendix A).

**Table 1 cancers-17-00358-t001:** Fitting parameters of the concentration-response relationships of KATP channel inhibitors in DIPG-36 cell at different incubation times.

EXP CONDITION	REPA CV			REPA CCK-8			GLIB CCK-8		
	IC_50_	HILL	Emax	IC_50_	HILL	Emax	IC_50_	HILL	Emax
DIPG-36 48 h	3.22 × 10^−6^ M	−4.17 slope	−100%	1.42 × 10^−4^ M	−1.57 slope	−58.78%	1.47 × 10^−4^ M	−1.78 slope	−51.10%
DIPG-36 72 h	3.22 × 10^−6^ M	−4.75 slope	−100%	8.57 × 10^−5^ M	−2.69 slope	−90.67%	9.47 × 10^−5^ M	−2.59 slope	−100%

REPA, repaglinide; GLIB, glibenclamide; IC_50_, molar concentration needed to reduce cell proliferation by 50%; Hill slope, slope of the concentration response relationships; Emax% percentage maximal reduction in the cell proliferation; CV, multi-wells crystal violet assay; CCK-8, mitochondrial dehydrogenase activity assay.

**Table 2 cancers-17-00358-t002:** Fitting parameters of the concentration-response relationships of KATP channel inhibitors in DIPG-50 cells at different incubation times.

EXP CONDITION	REPA CV			REPA CCK-8			GLIB CCK-8			GLIB CV		
	IC_50_	HILL	Emax	IC_50_	HILL	Emax	IC_50_	HILL	Emax	IC_50_	HILL	Emax
DIPG-36 48 h	1.27 × 10^−4^ M	−5.1 slope	−100%	3.221 × 10^−6^ M	−4.165 slope	−100%	1.31 × 10^−7^ M	−0.41 slope	−100%	1.77 × 10^−4^ M	−3.1 slope	−100%
DIPG-36 72 h	1.76 × 10^−7^ M	−3.65 slope	−100%	1.29 × 10^−7^ M	−3.9 slope	−100%	8.29 × 10^−5^ M	−4.4 slope	−100%	1.9 × 10^−7^ M	−3.8 slope	−100

REPA, repaglinide; GLIB, glibenclamide; IC_50_, molar concentration needed to reduce cell proliferation by 50%; Hill slope, slope of the concentration response relationships, Emax% percentage maximal reduction in the cell proliferation; CV, multi-wells crystal violet assay; CCK-8, mitochondrial dehydrogenase activity assay.

## Data Availability

The original contributions presented in this study are included in the article/Appendix A. Further inquiries can be directed to the corresponding authors.

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
