# Peer review of "KATP Channel Inhibitors Reduce Cell Proliferation Through Upregulation of H3K27ac in Diffuse Intrinsic Pontine Glioma: A Functional Expression Investigation"

_cancers, 2025, doi:10.3390/cancers17030358_

Round 1

Reviewer 1 Report (Previous Reviewer 2)

Comments and Suggestions for Authors

Not all the comments made earlier have been addressed in this resubmitted version. The authors are requested to provide a response against each of the previous comments and how they have/ haven't been addressed in this resubmission.

Author Response

Rev1: Please divide the Abstract into the usual subtitles: Background, Methods, Results, and Conclusions. In addition, the Conclusions of the article should be clearer. The statement "antidiabetic therapy" is probably not very good.

Authors: Thank you for your favorable comments. The manuscript was revised accordingly, and new data were added to the file.

Reviewer 2 Report (Previous Reviewer 3)

Comments and Suggestions for Authors

The authors correctly improved the manuscript.

Author Response

Rev2: This manuscript by Antonacci et al. explores the sensitivity of Diffuse Intrinsic Pontine Glioma (DIPG) cells to KATP channel inhibitors. Though the study is interesting it is missing crucial controls. The experiments are not robust and do not fully support their claims. Additionally, the poor English and numerous grammatical errors make it very difficult to understand the manuscript. My comments are given below:

  1. The Y-axis in Fig 1B and 1C does not make sense. What do the negative values on the Y-axis indicate? The normalization should be done concerning ‘CTRL’ and plotted starting from 0, which denotes no cell survival. Same for Fig 6.

Authors: This is right, figures 1 and 6 were revised as requested. These changes can be found on page 6 fig 1 and page 11 fig 6.

  1. The authors claim that after Capsaicin treatment, the surviving fraction is 104.8±0.05 %. Fig 2C graph doesn’t match these numbers.

Authors: Sorry for the error, the scaling is wrong, we revised figure 2C accordingly. This change can be found on page 7 fig 2C.

  1. Glib CV missing from Fig 3. Difficult to compare efficacy of crystal violet (CV) and CCK-8 readouts for Glib treatment.

Authors: It is right but the data were not available, and the relative paragraph was reduced. This change can be found on page 8 lines 313, 316.

  1. The authors should perform clonogenic assays for DIPG-50 cell line. They claim that these cells are relatively non-adherent. In that case Matrigel or soft agar can be used.

Authors: It is right, thanks for the precious suggestion, as requested clonogenic assay using Matrigel was performed and the new data were included in the revised vs of the manuscript. This change can be found on page 11-12 fig 6D line 431.

  1. Figure 8 is missing.

Authors: sorry we introduced the requested figure.

  1. Figure 7 only confirms changes in cell size. To confirm cell death authors must either show Annexin V-AAD staining by flow cytometry or cleaved caspase-3/ cleaved PARP-1 by western blot.

Authors: as requested flow cytometry was performed for apoptosis evaluation and the new data were included in the revised vs of the manuscript. This change can be found on page 12, fig 7 line 437.

  1. Fig 10, unclear what is the control that has been used for plotting relative expression of these genes. What does the Y-axis represent? Is it a fold change with respect to β-actin? Do these genes have expression greater than actin in these cells?

Authors: Beta-actin is normally used as a housekeeping gene so our genes were indeed upregulated

  1. Proper controls missing from all experiments. This paper claims that ‘KATP channel inhibitors reduce cell proliferation through up-regulation of H3K27ac in Diffuse-intrinsic-pontine-glioma’. The authors have used only two cell lines, DIPG-36 and DIPG-50 to support their claim. However, they need to include control DIPG cells that don’t have the H3K27M mutation to show that this effect is H3K27M specific.

Authors: This is right but unfortunately neuronal human control cells from these patients are not easily available, and the cell culture is not so easy to grow. We however performed new cell proliferation experiments on murine bone marrow cells showing no significant changes in cell proliferation with repaglinide and glibenclamide after 72 hours of incubation time. The new data were now included in the revised vs of the manuscript. This change can be found in Supplementary files fig 2S line 486.

  1. H3K27M mutation is known to cause low H3K27me3 and high H3K27ac. What is the status of H3K27me3 after treatment with REPA?

Authors: we investigated the role of the Tri-Methyl H3(Lys27) which is also a key epigenetic element regulating histone, but no significant changes were observed with different concentrations of REPA in DIPG cells. The data were now included in the revised vs of the manuscript. This change can be found on page 16, Supplementary fig 3S, 4S,  line 450.

  1. Is there any change in expression of SUR2 or KATP channel genes after treatment with REPA in DIPG-36 and DIPG-50?

Authors: western blotting experiments of SUR2 were performed and the protein contents were not affected by treatments as reported in the revised vs of the manuscript. This change can be found on page 16, Supplementary fig 5S line 559.

  1. Is there any change in H3K27ac at the promoter of SUR2 or KATP channel genes after treatment with REPA?

Authors: See point 10.

  1. Figure 12A, the graph shows no significant change in total caspase-3 after treatment with REPA in DIPG-36. This does not match with the western blot that shows a substantial increase in total caspase-3.

Authors: Yes this is true, we revised the figure as indicated and we thank the reviewer for this indication. This change can be found on page 17, fig 11.

  1. Figure 12 needs to be supplemented with Annexin V-AAD flow to confirm cell death.       Authors: as requested flow cytometry however using caspase 3/7 was performed for apoptosis evaluation and the data were included in the revised vs of the manuscript. See above point 5.    
  2. It is unclear why the authors started studying signaling changes in the mTOR/Akt pathway. Rather they should focus on trying to understand the role of H3K27ac in regulating sensitivity to REPA? Is increase in H3K27ac an effect or cause of this sensitivity? More validation is needed to support their claims.

Authors: The reviewer is right, so in the revised vs of the manuscript we reduced all sections related to ERK and AKT in the result and discussion (see page 18, line 619), and we focused our attention on the role of the H3K27ac. See above point 9.

Overall recommendation: Reject

Round 2

Reviewer 1 Report (Previous Reviewer 2)

Comments and Suggestions for Authors

All the comments have been satisfactorily addressed.

This manuscript is a resubmission of an earlier submission. The following is a list of the peer review reports and author responses from that submission.

Round 1

Reviewer 1 Report

Comments and Suggestions for Authors

Please, divide Abstract into usual subtitles as Background, Methods, Results and Conclusions. In addition Filald Conclusions of the article should be more clear. Steatment "antidiabetic therapy" is probably not very much good.

Reviewer 2 Report

Comments and Suggestions for Authors

This manuscript by Antonacci et al. explore the sensitivity of Diffuse Intrinsic Pontine Glioma (DIPG) cells to KATP channel inhibitors. Though the study is interesting it is missing crucial controls. The experiments are not robust and do not fully support their claims. Additionally, the poor English and numerous grammatical errors makes it very difficult to understand the manuscript. My comments are given below:

1.      The Y-axis in Fig 1B and 1C does not make sense. What do the negative values on the Y-axis indicate? The normalization should be done with respect to ‘CTRL’ and plotted starting from 0, which denotes no cell survival. Same for Fig 6.

2.      The authors claim that after Capsaicin treatment, surviving fraction is 104.8±0.05 %. Fig 2C graph doesn’t match these numbers.

3.      Glib CV missing from Fig 3. Difficult to compare efficacy of crystal violet (CV) and CCK-8 readouts for Glib treatment.

4.      The authors should perform clonogenic assays for DIPG-50 cell line. They claim that these cells are relatively non-adherent. In that case Matrigel or soft agar can be used.

5.      Figure 8 is missing.

6.      Figure 7 only confirms changes in cell size. To confirm cell death authors must either show Annexin V-AAD staining by flow cytometry or cleaved caspase-3/ cleaved PARP-1 by western blot.

7.      Fig 10, unclear what is the control that has been used for plotting relative expression of these genes. What does the Y-axis represent? Is it a fold change with respect to β-actin? Do these genes have expression greater than actin in these cells?

8.      Proper controls missing from all experiments. This paper claims that ‘KATP channel inhibitors reduce cell proliferation through up-regulation of H3K27ac in Diffuse-intrinsic-pontine-glioma’. The authors have used only two cell lines, DIPG-36 and DIPG-50 to support their claim. However, they need to include control DIPG cells that don’t have the H3K27M mutation to show that this effect is H3K27M specific.

9.      H3K27M mutation is known to cause low H3K27me3 and high H3K27ac. What is the status of H3K27me3 after treatment with REPA?

10.   Is there any change in expression of SUR2 or KATP channel genes after treatment with REPA in DIPG-36 and DIPG-50?

11.   Is there any change in H3K27ac at the promoter of SUR2 or KATP channel genes after treatment with REPA?

12.   Figure 12A, the graph shows no significant change in total caspase-3 after treatment with REPA in DIPG-36. This does not match with the western blot that shows a substantial increase in total caspase-3.

13.   Figure 12 needs to be supplemented with Annexin V-AAD flow to confirm cell death.

14.   It is unclear why the authors start studying signaling changes in the mTOR/Akt pathway. Rather they should focus on trying to understand the role of H3K27ac in regulating sensitivity to REPA? Is increase in H3K27ac an effect or cause of this sensitivity? More validation is needed to support their claims.

Overall recommendation: Reject

Comments on the Quality of English Language

Numerous grammatical errors are present throughout the manuscript that must be addressed.

Reviewer 3 Report

Comments and Suggestions for Authors

In this manuscript, the authors demonstrated the antiproliferative effect of KATP  and TRPV1 channel inhibitors on DIPG-36 and DIPG-50 cell lines. Moreover, they provided evidence about  the presence of KATP 654 and TRPV1 channel currents in these cells and suggested a combination of therapy with the channel inhibitors repaglinide or glibenclamide in DIPG cells.

The manuscript is quite well written, nevertheless some improvements are needed:

Fig. 2: data obtained in the clonogenic assay are not significant. The experiment should be repeated almost three times in triplicate.

Why did you test capsaicin in the clonogenic assay?

Fig. 4 A, B, C, D: scale bars in the images are missing

Fig 5 A, B and Fig. 9:  in the graphs the significativity is missing

Fig. 11: the expression of H3K27-ac should be normalized on H3K27 expression rather then on the beta actin expression

line 573: activated Akt is not on Ser 2448 (as in p-mTOR) but on Ser 473

line 577: p42 is ERK1 and p44 is ERK2

Data on AKT and ERK activation should be shown, otherwise the description and the discussion of these results should be reduced.

Comments on the Quality of English Language

The quality of English language need to be improved